# *PTRH2* Gene Variants: Recent Review of the Phenotypic Features and Their Bioinformatics Analysis

**DOI:** 10.3390/genes14051031

**Published:** 2023-04-30

**Authors:** Rajech Sharkia, Sahil Jain, Muhammad Mahajnah, Clair Habib, Abdussalam Azem, Wasif Al-Shareef, Abdelnaser Zalan

**Affiliations:** 1Unit of Human Biology and Genetics, Triangle Regional Research and Development Center, Kfar Qari 30075, Israel; w.shareef@gmail.com (W.A.-S.); dr.zalan@hotmail.com (A.Z.); 2Unit of Natural Sciences, Beit-Berl Academic College, Beit-Berl 4490500, Israel; 3Department of Biochemistry and Molecular Biology, Faculty of Life Sciences, Tel Aviv University, Tel Aviv 69978, Israel; sahilj@tauex.tau.ac.il (S.J.); azema@tauex.tau.ac.il (A.A.); 4The Ruth and Bruce Rappaport Faculty of Medicine, Technion-Israel Institute of Technology, Haifa 31096, Israel; muhamadmah@hymc.gov.il; 5Child Neurology and Development Center, Hillel Yaffe Medical Center, Hadera 38100, Israel; 6Genetics Institute, Rambam Health Care Campus, Haifa 31096, Israel; clair.habib@rch.org.au

**Keywords:** autosomal recessive disorder, bioinformatics analysis, clinical features, IMNEPD, *PTRH2* gene, PTRH2 variants, rare genetic diseases

## Abstract

Peptidyl-tRNA hydrolase 2 (PTRH2) is an evolutionarily highly conserved mitochondrial protein. The biallelic mutations in the *PTRH2* gene have been suggested to cause a rare autosomal recessive disorder characterized by an infantile-onset multisystem neurologic endocrine and pancreatic disease (IMNEPD). Patients with IMNEPD present varying clinical manifestations, including global developmental delay associated with microcephaly, growth retardation, progressive ataxia, distal muscle weakness with ankle contractures, demyelinating sensorimotor neuropathy, sensorineural hearing loss, and abnormalities of thyroid, pancreas, and liver. In the current study, we conducted an extensive literature review with an emphasis on the variable clinical spectrum and genotypes in patients. Additionally, we reported on a new case with a previously documented mutation. A bioinformatics analysis of the various *PTRH2* gene variants was also carried out from a structural perspective. It appears that the most common clinical characteristics among all patients include motor delay (92%), neuropathy (90%), distal weakness (86.4%), intellectual disability (84%), hearing impairment (80%), ataxia (79%), and deformity of head and face (~70%). The less common characteristics include hand deformity (64%), cerebellar atrophy/hypoplasia (47%), and pancreatic abnormality (35%), while the least common appear to be diabetes mellitus (~30%), liver abnormality (~22%), and hypothyroidism (16%). Three missense mutations were revealed in the *PTRH2* gene, the most common one being Q85P, which was shared by four different Arab communities and was presented in our new case. Moreover, four different nonsense mutations in the *PTRH2* gene were detected. It may be concluded that disease severity depends on the *PTRH2* gene variant, as most of the clinical features are manifested by nonsense mutations, while only the common features are presented by missense mutations. A bioinformatics analysis of the various *PTRH2* gene variants also suggested the mutations to be deleterious, as they seem to disrupt the structural confirmation of the enzyme, leading to loss of stability and functionality.

## 1. Introduction

Infantile-onset multisystem neurologic, endocrine, and pancreatic disease (IMNEPD) is a very rare autosomal recessive disorder first described in 2014 [1]. Studies found that this disorder was caused by biallelic mutations in the peptidyl-tRNA hydrolase 2 (*PTRH2*) gene located on chromosome 17 (NG_042064.1) [2,3]. Since it is a multisystem disorder, it may result in several variable phenotypes. The clinical features of IMNEPD include global developmental delay (with motor and speech delay), intellectual disability, sensorineural hearing loss, ataxia, pancreatic insufficiency (both exocrine and endocrine), postnatal microcephaly, peripheral neuropathy, facial dysmorphism, cerebellar atrophy, hypothyroidism, diabetes mellitus, and liver dysfunction [3,4,5,6].

PTRH2, also known as BIT1 (Bcl-2 inhibitor of transcription 1), is an evolutionarily, highly conserved protein that belongs to a family of peptidyl tRNA hydrolases. It releases the peptidyl moiety from tRNA, thus preventing the accumulation of prematurely dissociated peptidyl-tRNA, which could inhibit protein synthesis and be toxic to cells. Furthermore, the PTRH2 protein plays a role in regulating cell survival and death. It promotes cell survival and regulates muscle differentiation in human development. On the other hand, loss of PTRH2 function due to gene mutations causes congenital IMNEPD [7,8,9].

Human PTHR2 is a 19 kDa protein present at multiple locations, including the mitochondrial outer membrane, plasma membrane, endoplasmic reticulum, and the Golgi apparatus [10,11,12], and is expressed in all tissue types [13,14]. It consists of 179 residues (UniProt ID: Q9Y3E5) and is suggested to participate in multiple localization-specific functions [15]. Some of its suggested functions include regulation of adhesion signals and B-cell lymphoma 2 (Bcl2) gene expression [15]; modulation of Phosphoinositide 3-kinase/Protein kinase B (PI3K/AKT) and extracellular signal-regulated kinase (ERK) signaling [15]; initiation of anoikis upon loss of adhesion [10,15]; regulation of myogenic differentiation [1,11,16,17]; regulation of mechanistic target of rapamycin (mTOR) pathway; and epithelial-to-mesenchymal transition (EMT) in neural development [18,19]. These functionalities may be affected by the integrin-mediated adhesion, binding co-factors, and phosphorylation state of the protein [15].

Mutations in the *PTRH2* gene can lead to a reduction or loss of function in the PTRH2 protein, which may disrupt the process of mitochondrial translation and contribute to the development of IMNEPD. As per previous reports, missense and nonsense mutations in the *PTRH2* gene may lead to IMNEPD cases exhibiting variable severity [20].

Availability of a vast array of computational tools and servers, including protein structure prediction servers (such as AlphaFold V_2), protein-binding site prediction servers (such as SPPIDER V_2 and ScanNet V_0.3), evolutionary conservation prediction servers (such as Consurf), protein stability measurement servers (such as Rossetta-PM), and post-translational modification prediction servers (such as Venus and MusiteDeep) facilitate gaining insight into the effects of mutations on the functionality of a protein from a structural perspective. For example, in a recent study, a protein bioinformatics analysis was reported for the *CLN8* gene [21]. In another computational study, Suleman et al. highlighted the structural stability conferred by mutations in the SARS-CoV-2 spike protein enabling the protein to bind with the ACE2 receptor [22].

In the current study, an extensive review was carried out for all the reported IMNEPD cases, along with their variable clinical spectrum in relation to their respective genotypes. We also reported on a new local case with a known *PTRH2* gene variation. Furthermore, we presented the relative frequencies of various clinical manifestations of all the cases. Additionally, a bioinformatics analysis of the various *PTRH2* gene variants was carried out from a structural perspective to provide insight into the effects of mutations on the structural-functional relationship of the enzyme.

## 2. Methodology

In the current study, all available cases with *PTRH2* gene variants (up until the date of preparation of this study, end of the year 2022) were enlisted with their clinical features (Table 1). The literature review had been carried out extensively using the PubMed and Google Scholar websites (https://pubmed.ncbi.nlm.nih.gov/, accessed on 10 February 2023 and https://scholar.google.com/, accessed on 10 February 2023). Additionally, the relative frequencies of various clinical manifestations and genetic variables in all these cases were summarized (Table 2).

### Structural Analysis of the Mutational Effects

For the structural analysis of the mutations, the three-dimensional structure of the PTRH2 catalytic hydrolase domain (residues 63–179) derived from crystallographic studies (PDB ID: IQ7S) was obtained. The structure of the N-terminal residues 1–62 was predicted with the help of AlphaFold [27]. AlphaFold is a modern neural network-based model that predicts protein structures with atomic accuracy and does not require a homolog structure. The confidence of prediction was confirmed by the predicted local-distance difference test (pLDDT) [27]. ConSurf webserver [28] was used to predict the evolutionary conservation of various amino acids in the PTRH2 enzyme. A higher evolutionary conservation plausibly indicates the critical nature of the residue. ConSurf is trained to identify homologs to a query sequence and identify evolutionary relationship patterns based on multiple sequence alignment [28]. Furthermore, SPPIDER (Solvent accessibility-based Protein-Protein Interface iDEntification and Recognition) and ScanNet (Spatio-Chemical Arrangement of Neighbors Network) webservers were used to predict the protein-protein interaction sites. SPPIDER is a machine learning-based protein interface recognition server that utilizes relative solvent accessibility (RSA) predictions and high-resolution structural data to estimate the participation of various residues in protein-protein interactions [29]. ScanNet [30] is a multi-scale geometric deep learning (DL)-based server that predicts the involvement of a particular residue in structural motifs based on the 3-D conformation of its neighboring residues [30]. Rosetta-PM [31] was also employed to estimate the effect of mutations on the stability of the 3-D structure, while Venus [32] (https://venus.cmd.ox.ac.uk, accessed on 23 February 2023) and MusiteDeep [33] (https://www.musite.net, accessed on 23 February 2023) webservers were used to estimate the effect of mutations on post-translational modification sites. Venus is based on Michelanglo, while MuSiteDeep is a DL-based webserver facilitating efficient real-time prediction of post-translational modification sites for multiple sequences input. Additionally, PhosphoSite Plus, an interactive resource encompassing nearly 130,000 experimentally observed post-translational modification sites [34], was used to predict phosphorylation sites in the PTRH2 protein.

## 3. Results

### 3.1. Case Presentation

A family from the Arab community in Israel was referred to the Genetics Institute at the Rambam Health Care Campus, Haifa, since their child suffered from global developmental delay, sensorineural hearing loss, and peripheral neuropathy. The parents were second-cousin relatives with an affected son and two healthy siblings.

The mother’s pregnancy was reported as uneventful, although limited follow-up was performed, without any prenatal scans or genetics screening. The affected son was born at 35 weeks of gestational age via a normal vaginal delivery, and his birth weight was 2.3 kg. The infant required respiratory support after birth due to respiratory distress secondary to suspected congenital pneumonia. At seven months, he was diagnosed with progressive sensorineural hearing loss. He was initially assisted with hearing aids, followed by cochlear implants at two years of age. He had also experienced global developmental delay, indicated as follows: he sat well at the age of 1.5 years, walked at the age of three years but with gait difficulties, and started talking after the cochlear implant surgery. He was examined at the age of three years, and his physical and neurological exam revealed hypotonia with normal tendon reflexes, while EMG/NCV demonstrated demyelinating axonal polyneuropathy, both sensory and motoric.

At the age of 2.5 years, massive and foul-smelling stools were reported. Also, at the age of six years, the outcomes of stool elastase, along with Vitamin A and E deficiency tests, were indicative of exocrine pancreatic insufficiency. Subsequently, treatment with pancreatic enzymes was initiated.

At the age of seven years, a neuro-developmental evaluation revealed that he was still consistent with global delay; he still had gait difficulties and frequent falls, could not climb stairs without assistance, and could not jump. The patient also had grapho-motor difficulties in writing and drawing and attended a special education kindergarten. His most recent physical examination (at the age of seven years) revealed the following parameters: height 118 cm (25th centile), weight 22 kg (35th centile), and head circumference 49.5 cm (3rd centile). He showed additional mild dysmorphic features: hooded eyelids, down-slanting palpebral fissures, wide nasal bridge with mild epicanthal folds, low insert columella, prominent and pointed chin with prognathism, mild lower limb hypotonia with partial Gower sign, and mild pes cavus with hammer toes. However, no muscle atrophy was detected. The joint flexibility Beighton score was 6/9, and he presented with normal tendon reflexes.

The routine genetic evaluation, including chromosomal microarray analysis (CMA) was carried out, and no chromosomal microarray aberrations were detected. Since the clinical manifestations of the proband resembled those of IMNEPD, sequencing for the *PTRH2* gene was performed. The results showed that the patient is homozygous for the missense mutation p.Gln85Pro (c.254A > C), a pathogenic *PTRH2* gene variant identified previously by our group [3].

Since the first case of IMNEPD that was described in 2014, to date about 25 cases have been reported. In the current study, a systematic review of all these cases, along with their clinical manifestations, was carried out and is presented in Table 1. The relative frequencies of various clinical features of all the cases were also calculated and are shown in Table 2. It was found that the most common clinical characteristics among all the patients were motor delay (92%), neuropathy (90%), distal weakness (86.4%), intellectual disability (84%), hearing impairment (80%), ataxia (79%), and deformity of head and face (~70%). The lesser common characteristics included hand deformity (64%), cerebellar atrophy/hypoplasia (47%), and pancreatic abnormality (35%), while the least common characteristics were diabetes mellitus (~30%), liver abnormality (~22%), and hypothyroidism (16%). It is noteworthy that no clinical characteristic common to all the documented cases could be elucidated (Table 2).

The results revealed that the majority of the *PTRH2* gene variants detected in the 25 cases were missense mutations with a relative frequency of 64%, while the nonsense mutations showed a relative frequency of 36% (Table 2). Three missense mutations in the *PTRH2* gene were reported, the most common one being Q85P, which was reported in four different Arab communities from Saudi Arabia, Tunisia, Palestine, and the Arab community in Israel, as shown in Table 1. The relative frequency of the other two missense mutations (V23A and Y94N) was 4% each, and each of them was detected in only one separate case (Table 2).

Four different nonsense mutations were reported in the *PTRH2* gene. Three of them (A90Gfs*13, W108*, and E110*) were found in one family each, while the fourth (S43Kfs*11) was reported in two unrelated families (Table 1).

### 3.2. Structural Analysis of the Mutational Effects

The predicted 3-D structure of the PTRH2 protein is shown in Figure 1a. A visual inspection of the predicted structure indicates a largely unstructured N-terminus (except for residues 14–33), a small α helix, and a five-strand β-sheet flanked by two α-helices [9]. Residues 1–62 are suggested as being the mitochondrial localization sequence (MLS) [15]. A peptide fragment (residues 14–33) involved in cell death via anoikis was designated by Chen et al. as being the PTRH2-cell death domain (PTRH2-CDD) [35] (Figure 1a). Residues 63–179 are designated as the catalytic hydrolase domain [15] and are suggested to be involved in the cell survival role of PTRH2 [35]. Residues 80–99 forming the big α helix are considered to be involved in binding with various proteins [9,15] (Figure 1a).

Below we present a structural analysis of the mutations reported in the PTRH2 protein (UniProt ID: Q9Y3E5) in order to predict their functional importance.

#### 3.2.1. V23A

Judging by the 3-D protein structure (Figure 1a), V23 is part of the helical CDD in a largely unstructured N-terminus region. Such parts are often involved in protein-protein interactions. Indeed, V23 was predicted to be a protein-binding site by the SPPIDER [29] and ScanNet [30] webservers. This prediction is in support of the plausible significance of this residue in the regulation of cell apoptosis, as V23 mutation in the mitochondrial PTRH2 might result in the disruption of PTRH2-TLE (TLE: Transducin-like Enhancer of Split) interaction [15], consequently inhibiting cell death.

#### 3.2.2. S43Kfs*11

This mutation results in the formation of a truncated PTRH2 protein, which partially lacks MLS. Also, the truncated protein lacks the complete catalytic hydrolase domain, making it inactive.

#### 3.2.3. Q85P

Q85 is part of an electrostatic interaction network, which also involves K81, D145, and T157. This is a significant interaction network, as K81, Q85, and T157 are suggested to be part of a putative active site [36]. This network also stabilizes the protein and helps to maintain the correct position of helix 80–99 and the adjacent β-sheet, which together form the tight core of the protein. Within this interaction network, the side chain of Q85 forms a hydrogen bond with T157 and a polar hydrogen bond with D145 (Figure 2).

The fact that Q85 and T157 are suggested to be part of a putative active site [36] emphasizes the importance of the Q85-T157 interaction. Additionally, the Q85-D145 interaction might be important as it helps the correct positioning of D145 with respect to K81, allowing them to form a strong, stabilizing salt bridge (Figure 2). Furthermore, K81 is a plausible ubiquitylation site, as suggested by Akimov et al. [37] and as predicted by the Venus [32] and MusiteDeep [33] webservers. Therefore, the conformation of K81, due to its interaction with Q85 and D145, might be critical for exposing the ε-amino group of K81 to the C-terminus of ubiquitin, resulting in ubiquitination and, consequently, maintaining cell homeostasis. This conformation of K81 might change due to the change in electrostatic interactions between this residue, D145, and P85 following the mutation, thus disrupting homeostasis.

#### 3.2.4. A90Gfs*13

This mutation leads to the formation of a truncated, non-functional PTRH2 variant lacking the majority of the catalytic hydrolase domain.

#### 3.2.5. Y94N

Y94 is involved in various interactions with amino acids located on an adjacent short helix. These interactions include a hydrogen bond with E109, a weak π-π interaction with W108, and hydrophobic interactions with L105 (Figure 3).

This interaction network stabilizes the part of the protein that contains the two helices and also helps positioning the short helix in a way that allows W108, which is highly conserved, to form multiple interactions with other parts of the protein. These interactions include an electrostatic interaction with K115, π-π interactions with Q113, sulfur-π interactions with M67, and hydrophobic interactions with both M67 and P163 (Figure 3). Thus, Y94 is both directly and indirectly responsible for the stabilization of this part of the protein. Interestingly, Q113 and K115 are highly conserved residues, as indicated by the ConSurf analysis (Figure 1b), and were predicted to be protein-binding sites by the SPPIDER [29] and ScanNet [30] webservers. Thus, interaction with W108, which is itself a highly conserved residue, might be significant for the conformational positioning of Q113 and K115, enabling their interaction with other proteins. This notion is further supported by sequence alignment in the CDD/SPARCLE database [36], which suggests that K115 is part of a putative active site. Therefore, a Y94N mutation might have a detrimental effect on interactions between PTRH2 and other proteins.

#### 3.2.6. W108*

W108 is a stop mutation that results in the formation of a truncated PTRH2 protein without a major chunk of the catalytic hydrolase domain.

#### 3.2.7. E110*

E110 is a deleterious mutation that results in the formation of a truncated PTRH2 protein without a major chunk of the catalytic hydrolase domain.

## 4. Discussion

In this manuscript, we reported on a new case with a previously documented mutation. Furthermore, we endeavored to enlist previously reported phenotypic symptoms for each (missense and nonsense) condition and provide a computational insight into the plausible reasons for the severity of the symptoms observed. To this end, we performed a structural analysis to understand the effect of various mutations (missense and nonsense) on the intra-molecular stability and overall conformation of the protein. An analysis of the various nonsense and missense mutations in the PTRH2 enzyme suggested that, overall, the missense mutations result in less severe phenotypes compared to the nonsense mutations. This finding could be concluded from the observation that patients with missense mutations had less common clinical features of IMNEPD compared to those with nonsense mutations (Table 1). Furthermore, the bioinformatics analysis of the variants of PTRH2 suggested several mechanisms through which the point mutations could diminish the biological activity of the protein. Most of the missense variants were estimated to have a lesser protein-protein interaction ability, consequently disrupting various signaling processes. Additionally, most point mutations were suggested to reduce the structural stability of the enzyme by disrupting the intra-protein interaction network. Structural destabilization may influence functionality of the enzyme, resulting in the observed symptomatic conditions.

Specific to missense mutations, three mutations have been documented in the *PTRH2* gene, with the most common (Q85P; 56%) being reported in four different Arab communities (Saudi, Tunisian, Palestinian, and the Arab community in Israel). Interestingly, the Q85P mutation is present mainly in the Arab communities. This phenomenon could be explained by the common-founder effect, as this mutation probably originated in Saudi Arabia. The main distinguishing clinical manifestations for this mutation seem to be motor delay, intellectual disability, hearing impairment, neuropathy, deformity of head and face, distal weakness, hand deformity and ataxia. On the other hand, the less common clinical features for this mutation appear to be cerebellar atrophy and liver and pancreatic abnormality, while diabetes mellitus and hypothyroidism were not reported.

Based on the bioinformatics analysis, the Q85P mutation is expected to have two deleterious effects on enzyme function. The first is loss of catalytic activity, and the second is a decrease in overall stability of the protein due to loss of the side chain interactions of Q85 with the surrounding residues. Another factor that is expected to decrease the stability further due to the Q85P mutation is the imino-amino-to change of the backbone nitrogen of this residue, which leads to the loss of a hydrogen bond donor. Indeed, Rosetta-PM [31] calculations suggest that the Q85P mutation is likely to decrease protein stability by nearly 11 kcal/mol. Thus, the Q85P mutation in the PTRH2 enzyme may be considered as a milder phenotype of the disease compared to nonsense mutations.

The other two missense mutations, V23A and Y94N, had been documented in Iranian and Japanese ethnicities, respectively [20,25]. According to Khamirani et al. [25], the V23A mutation may result in movement disorders, motor delay, and severe myopia, though the gastrointestinal examination was unremarkable. Also, facial dysmorphism and hearing disorders were not reported. This absence could be due to the similar nature of valine and alanine residues; consequently, the damage resulting from the V23A mutation might exhibit only minimal effects.

The bioinformatics analysis also indicated that V23 is part of a PTRH2 N-terminal segment suggested to contain a mitochondrial localization sequence (MLS) [15] (Figure 1a). It is also part of the PTRH2-CDD, which is involved in cell death via anoikis [35] (Figure 1a). Therefore, a mutation at V23 might affect either the protein localization into the mitochondria or the caspase-independent cell apoptosis function of the protein, or both. Specifically, PTRH2 is suggested to anchor at the ER and Golgi membrane with the help of its N-terminus, with the C-terminal domain protruding to the cytoplasm [12,15]. Mutation at V23, present near the N-terminus, might prevent the binding of PTRH2 to the ER and Golgi surface. This variation might consequently disrupt the association of the PTRH2 C-terminal catalytic domain with the Extracellular Signal-Regulated Kinase 1 (ERK1) [12,15], leading to continuation of the ERK signaling. Also, disruption in the *PTRH2*-ERK1 complex formation may also lead to inhibition of anoikis in retinal astrocytes during the remodeling of the developing eye [15]. Overall, it may be suggested that V23A mutation leads to milder effects than Q85P and nonsense mutations.

The third missense mutation, Y94N, was described by Ando et al. [20] in a patient from Japan. The patient was diagnosed with hearing loss and mild mental retardation during childhood, though motor impairment, cognitive decline, and gait instability were not documented until the age of 50 years. Furthermore, his neurological symptoms included distal dominant muscle weakness and atrophy, reduced tendon reflex, foot deformity without sensory symptoms, limb ataxia, and truncal ataxia. Multiple small malformations, severe axonal sensorimotor neuropathy, and cerebellar hypoplasia were also reported.

The bioinformatics analysis indicated that Y94 is part of the PTRH2 catalytic domain. The Y94N mutation is expected to cause the loss of significant interactions, thereby destabilizing this part of the protein (Figure 3). Indeed, Rosetta-PM [31] suggests that the mutation is likely to lead to a 3.6 kcal/mol destabilization of the protein. Additionally, as suggested by Possemato et al. [38] and predicted by Venus [32] and PhosphoSite Plus [34] webservers, Y94 is a plausible phosphorylation site. Therefore, Y94 might be significant for the cell survival function of the PTHR2 enzyme. Moreover, Y94 is part of the 80–99 big α helix, which is considered to act as a protein-binding site [9,15]. Indeed, SPPIDER [29] and ScanNet [30] webservers indicate a plausible protein-binding site at Y94. Therefore, a Y94N mutation might have a detrimental effect on interactions between *PTRH2* and other proteins. Overall, in comparison to the other two missense mutations (Q85P, V23A), the Y94N mutation seems to be more severe, based on the clinical manifestations (Table 1). Therefore, we can expect nonsense mutations to be the most detrimental, followed by Y94N, then Q85P, and last, the mildest mutation, V23A.

Our literature review indicated that the nonsense mutations S43Kfs*11, A90Gfs*13, W108*, and E110* might lead to a non-functional PTRH2 protein lacking the majority of or a complete catalytic hydrolase domain. These mutations may be suggested to result in a more severe phenotype of the IMNEPD characterized by the most common clinical features (intellectual disability, motor delay, neuropathy, distal weakness, ataxia, hearing impairment, and deformity of head and face), as well as the less common clinical phenotypes (hand deformity, cerebellar atrophy/hypoplasia, diabetes mellitus, pancreatic and liver abnormalities, and hypothyroidism). In a recent study, one case with a nonsense mutation in the PTRH2 protein (S43Kfs*11) was suggested to present with a milder phenotype compared to the cases with missense mutations [5]. However, a later study by Becker et al. [26] described two patients with the S43Kfs*11 mutation exhibiting a more severe phenotype than reported earlier. In addition to these contrasting reports, phenotypic variabilities between siblings and among various cases with the same mutation were also observed. A possible explanation for these phenotypic variabilities could be the variable expressivity of the *PTRH2* gene mutations.

PTRH2 has been suggested to play a role in various cellular functions that include regulation of adhesion signals, B-cell lymphoma 2 (Bcl2) gene expression [15], modulation of Phosphoinositide 3-kinase/Protein kinase B (PI3K/AKT), extracellular signal-regulated kinase (ERK) signaling [15], initiation of anoikis upon loss of adhesion [10,15], regulation of myogenic differentiation [1,11,16,17], regulation of mechanistic target of the rapamycin (mTOR) pathway [1], and epithelial-to-mesenchymal transition (EMT) in neural development [18,19]. These functionalities may be affected by the integrin-mediated-adhesion, binding co-factors, and phosphorylation state of the protein [15]. The effect of various PTRH2 mutations on these functional pathways remains to be elucidated. An understanding of these effects could help gain insight into the development of consequential multiple manifestations observed in IMNEPD patients.

It is evident that IMNEPD involves many systems in the patients that could lead to a wide differential diagnosis. Thus, different patients may have overlapping clinical characteristics with various syndromes and metabolic diseases, such as Johanson-Blizzard syndrome and MELAS (mitochondrial encephalomyopathy, lactic acidosis, and stroke-like episodes). As the wide spectrum of symptoms and associated diseases make the diagnosis of this disease particularly challenging, we suggest detecting mutations in the *PTRH2* gene during clinical and genetic diagnosis in the relevant health centers by the concerned health professionals as an indicator for IMNEPD.

## Figures and Tables

**Figure 1 genes-14-01031-f001:**
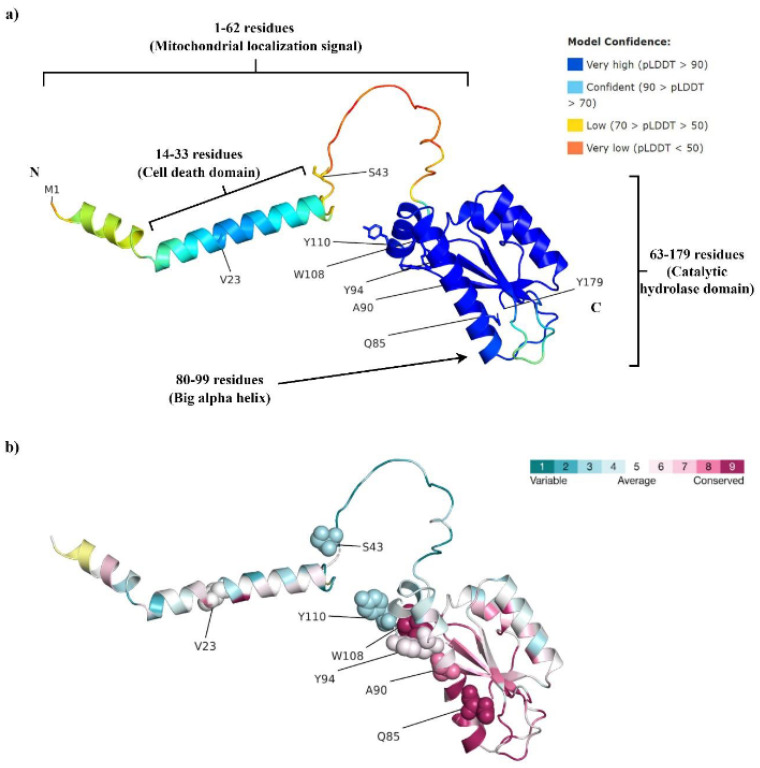
(**a**) The AlphaFold-predicted structure of human *PTRH2*. Residues 1–62 are suggested to be the mitochondrial localization signal, residues 14–33 are known as the cell death domain, while residues 63–179 are the catalytic hydrolase domain. Residues 80–99, forming the big α helix, are thought to be involved in binding with various proteins. (**b**) Consurf analysis indicates a high evolutionary conservation of Q85, W108, and A90, while average conservancy was predicted for Y94 and V23.

**Figure 2 genes-14-01031-f002:**
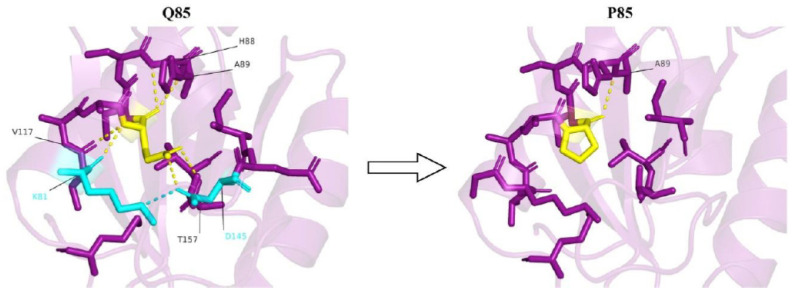
A strong electrostatic interaction network is observed in the presence of glutamine at the 85th position (yellow). Q85 forms polar interactions (yellow dashes) with K81, H88, A89, V117, D145, and T157. Most of these interactions are lost upon Q85P mutation. Also, due to the conformational position of D145 owing to the interaction with Q85, D145 is able to form a salt bridge (cyan dash) with K81. No such salt bridge formation is observed upon Q85P mutation.

**Figure 3 genes-14-01031-f003:**
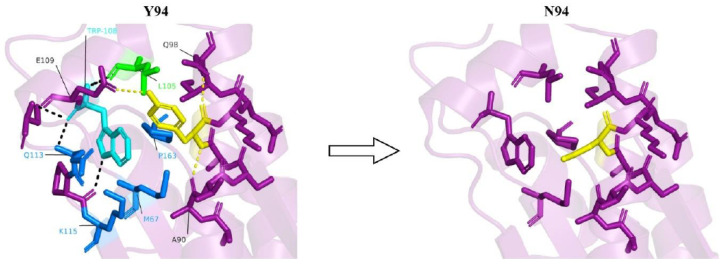
In the presence of tyrosine at the 94th position (yellow), a host of interactions, including hydrogen bonds with A90, Q98 and E109 (yellow dashes), weak π-π interaction with W108 (cyan residue), and hydrophobic interaction with L105 (green residue), are observed. Due to the conformational positioning of W108 owing to its π-π interaction with Y94, it is able to interact with multiple residues (black dashes and blue residues). No such interactions are observed upon Y94N mutation.

**Table 1 genes-14-01031-t001:** Phenotypic characteristics and *PTRH2* gene variants of all the IMNEPD published cases including our current case.

Mutation Type	Missense Mutations							Nonsense Mutations				
*PTRH2* variant ^a^	c.254A > C					c.68 T > C	c.280 T > A	c.269_270delCT	c.324G > A	c.127dupA		c.328G > T
*PTRH2* protein variant	p.Gln85Pro **(Q85P)**					p.Val23Ala **(V23A)**	p.Tyr94Asn **(Y94N)**	p.Ala90Glyfs*13 **(A90Gfs*13)**	p.Trp108* **(W108*)**	p.Ser43Lysfs*11 **(S43Kfs*11)**		p.Glu110* **(E110*)**
Reference	[23]	[2]	[3]	[24]	Current Case	[25]	[20]	[1]	[4]	[5]	[26]	[6]
Ethnicity	Arab	Arab	Arab	Arab	Arab	Iranian	Japanese	Turkish	Arab	Indian	NR ^b^	Indian
Number of patients	1	5	3	4	1	1	1	2	3	1	2	1
**Clinical Features**												
Motor delay	14/14					1/1	0/1	2/2	3/3	0/1	2/2	1/1
Intellectual disability	11/14					0/1	1/1	2/2	3/3	1/1	2/2	1/1
Hearing impairment	10/14					0/1	1/1	2/2	1/3	1/1	2/2	1/1
Deformity of head and face	10/14					0/1	1/1	2/2	2/3	0/1	NR	0/1
Hand deformity	10/14					0/1	1/1	2/2	1/3	0/1	2/2	0/1
Distal weakness	11/13 ^c^					1/1	1/1	2/2	2/3	0/1	NR	1/1
Ataxia	6/9					NR	1/1	2/2	3/3	0/1	2/2	1/1
Cerebellar atrophy/hypo-plasia	2/9					NR	1/1	2/2	2/3	0/1	NR	1/1
Neuropathy	8/9					0/1	1/1	2/2	3/3	1/1	2/2	1/1
Liver abnormality	1/9					0/1	0/1	2/2	0/3	0/1	NR	1/1
Pancreatic abnormality	1/9					0/1	1/1	1/2	2/3	0/1	1/2	1/1
Hypothyroid-ism	0/14					0/1	1/1	2/2	0/3	0/1	1/2	0/1
Diabetes mellitus	0/13					NR	0/1	2/2	2/3	1/1	1/2	1/1

^a^ All the variants were homozygous. ^b^ NR: not reported. ^c^ When a particular clinical feature is not mentioned in the article for that specific patient, then that patient is excluded from the rest of the patients; therefore, the total number of patients will be reduced.

**Table 2 genes-14-01031-t002:** Frequency of various clinical and genetic variables of the IMNEPD reported cases.

Factors	Relative Frequency (N = 25)
Total number of reported families	14
**Number of patients affected by:**	
**Missense mutations:**	16 (64%)
(A) c.254A > C, p.Gln85Pro	14 (56%)
(B) c.68T > C, p.Val23Ala	1 (4%)
(C) c.280T > A, p.Tyr94Asn	1 (4%)
**Nonsense mutations:**	9 (36%)
(A) Nonsense point mutation (c.324G > A, p.Trp108*, c.328G > T, p.Glu110*)	4 (16%)
(B) Nonsense nucleotide deletion (c.269_270delCT, p.Ala90Glyfs*13)	2 (8%)
(C) Nonsense nucleotide duplication (c.127dupA, p.Ser43Lysfs*11)	3 (12%)
Motor delay	23/25 (92%)
Intellectual disability	21/25 (84%)
Hearing impairment	20/25 (80%)
Deformity of head and face	16/23 ^c^ (69.5%)
Hand deformity	16/25 (64%)
Distal weakness	19/22 (86.4%)
Ataxia	15/19 (79%)
Cerebellar atrophy/hypoplasia	8/17 (47%)
Neuropathy	18/20 (90%)
Liver abnormality	4/18 (22.2%)
Pancreatic abnormality	7/20 (35%)
Hypothyroidism	4/25 (16%)
Diabetes mellitus	7/23 (30.4%)

^c^ When a particular clinical feature is not mentioned in the article for that specific patient, then that patient is excluded from the rest of the patients; therefore, the total number of patients will be reduced.

## Data Availability

Not applicable.

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
