# Peer review of "PTRH2 Gene Variants: Recent Review of the Phenotypic Features and Their Bioinformatics Analysis"

_genes, 2023, doi:10.3390/genes14051031_

Round 1

Reviewer 1 Report

This manuscript collected information from previously published results as well as one patient they identified to determine if there is a correlation between phenotype and genotype of mutations in PTRH2 that causes IMNEPD. In addition, the authors used bioinformatic analyses to predict the functional consequences of missense mutations in the protein.

The major issue of this manuscript is the presentation of the data in tables 1 and 2. There is redundant information but still not the correct information. In addition, the actual tables could be cleaned up. More specifically, Tables 1 and 2 can be merged as they show very similar information. The columns in Table 1 should show missense mutations in a cluster and nonsense changes in another cluster. The percentage of each mutation should be listed in a row under the specific mutations and patient number. Also, the total number of patients and percentage with missense mutations that have a particular feature and the same for nonsense patients with a particular feature as well as the information for total numbers and percentages can be added as columns in table 1. Or if the authors want to compare each missense mutation separately that would be okay but percentages should be shown with the absolute numbers.

To make Table 1 easier to read, the authors can just use either the nucleotide change or protein change or if the authors want to list both, put them on different lines. The protein changes do not need to listed two different ways. Instead of writing homozygous for every mutation, that can be noted in a footnote in the table. The references in the table can be indicated by number and the first authors can be removed. For the ethnicity, the authors should just list ethnicity and not geographic location. For example, someone from Saudi Arabia is considered Arab but Syrians, Palestinians and Tunisians are also Arab. For Indians should they be considered South Asians? If geographic location is important than that should be listed on a separate line. Line 100 and 101, I do not understand the **footnote. I probably would not use the word deformity in the description of clinical features.

There are some issues in the discussion as well.

It is unclear for the genotype-phenotype correlation are the authors comparing all missense mutations as a group against all nonsense mutations? Or is each missense mutation being compared against all nonsense mutations or is each missense change compared to each other? Or all of the above? It’s very unclear and should be clarified.

One conclusion that the authors make is on Lines 284-287:

“An analysis of the various nonsense and missense mutations in the PTRH2 enzyme 284 suggested that the missense mutations result in less severe phenotypes. This could be concluded from observing the less common clinical features in patients with missense mutations, as compared to the most common features.” Do the authors mean that individuals with missense changes have smaller numbers of features than individuals with nonsense mutations? Does if matter if it’s a common or less common feature? To support this the authors should present their data as stated above.

Then they specifically discuss the Q85P mutation and list some “distinguishing” features though the features they list all are found in individuals with other types of mutations. They need to state the caveat of the other missense mutations is that there is only 1 patient that they base their observations in the discussion.

The authors go into great lengths to discuss the possible consequences to the protein of each missense change but it seems they make a determination of which is the most deleterious by the phenotype and not so much based on the proposed functional consequences. This is likely because the protein is found in many cellular locations and has many functions and it is unknown the functions associated with each of the protein domains.

Other smaller issues include the following:

There are many instances that the word choices the authors use is confusing. One example is on line 37  and the word “derogatory”.

Line 72: Does mutations in PTRH2 disrupt translation in the mitochondria or general protein translation?

It is unclear why the tables are shown in the methods section when they should be relocated to the results section. The same with the case presentation.

The authors discuss variable expressivity of the S43Kfs*11 mutation but it could also be said for the Q85P and W108* mutations as well. They do not discuss why this variable expressivity might occur.

Once an acronym such as IMNEPD is identified, it should be used consistently throughout the manuscript. The authors define acronyms multiple times.

The results are subdivided and section 3.1 is indicated to be the bioinformatic analysis but prior to that the authors discuss the predicted protein structure which is also bioinformatically analyzed. The term bioinformatic analysis is also not very descriptive.

Author Response

Reviewer 1:

Thank you for your constructive review of our manuscript. All changes you requested or suggested (by you and other reviewers) had been incorporated in the manuscript and highlighted in yellow.

This manuscript collected information from previously published results as well as one patient they identified to determine if there is a correlation between phenotype and genotype of mutations in PTRH2 that causes IMNEPD. In addition, the authors used bioinformatic analyses to predict the functional consequences of missense mutations in the protein.

The major issue of this manuscript is the presentation of the data in tables 1 and 2. There is redundant information but still not the correct information. In addition, the actual tables could be cleaned up. More specifically, Tables 1 and 2 can be merged as they show very similar information. The columns in Table 1 should show missense mutations in a cluster and nonsense changes in another cluster. The percentage of each mutation should be listed in a row under the specific mutations and patient number. Also, the total number of patients and percentage with missense mutations that have a particular feature and the same for nonsense patients with a particular feature as well as the information for total numbers and percentages can be added as columns in table 1. Or if the authors want to compare each missense mutation separately that would be okay but percentages should be shown with the absolute numbers.

Response:

  • We redesigned table 1 in a manner to look clearer and more informative as we divided all the variants into the two main mutations (i.e., the missense and the nonsense mutations) as you requested. We tried merging the two tables together, but this resulted into a more complex table in which it would be difficult for the readers to correlate the data together and it would seem very compacted. This would also cause difficulty in explaining and describing the relevant data in the text of the results. Additionally, one of the other reviewers recommended the split of the first table into two tables as it looks too congested with information.
  • The information presented in both tables were again verified and ensured to be correct and accurate as published in the literature.

To make Table 1 easier to read, the authors can just use either the nucleotide change or protein change or if the authors want to list both, put them on different lines. The protein changes do not need to listed two different ways. Instead of writing homozygous for every mutation, that can be noted in a footnote in the table. The references in the table can be indicated by number and the first authors can be removed. For the ethnicity, the authors should just list ethnicity and not geographic location. For example, someone from Saudi Arabia is considered Arab but Syrians, Palestinians and Tunisians are also Arab. For Indians should they be considered South Asians? If geographic location is important than that should be listed on a separate line. Line 100 and 101, I do not understand the **footnote. I probably would not use the word deformity in the description of clinical features.

Response:

All of your suggestions regarding the modifications of table 1 have been performed, except that we kept displaying the protein changes in two different ways while the second expression was extensively used in the text during the process of bioinformatics analysis as generally adopted.

There are some issues in the discussion as well.

 It is unclear for the genotype-phenotype correlation are the authors comparing all missense mutations as a group against all nonsense mutations? Or is each missense mutation being compared against all nonsense mutations or is each missense change compared to each other? Or all of the above? It’s very unclear and should be clarified.

Response:

The comparison includes all the above options. The necessary modifications and clarifications have been undertaken wherever appropriate. 

In this manuscript, we have endeavored to enlist the previously reported phenotypic symptoms for each (missense and nonsense) condition and provide a computational insight into the plausible reasons for the severity of the observed symptoms. In this direction, we performed a structural analysis to understand the effect of various mutations (missense and nonsense) on the intra-molecular stability and overall conformation of the protein. Then, in the discussion section, we tried to contemplate the relation between the reported severity and the computationally expected changes in the molecular structure and consequently, the molecular function.

The same has now been clarified in the discussion at various locations. Some examples are as follows:

“…. overall, the missense mutations result in less severe phenotypes as compared to the nonsense mutations.”

“Q85P mutation in PTRH2 enzyme may be considered as a milder phenotype of the disease as compared to nonsense mutations.”

“Overall, it may be suggested that V23A mutation leads to milder effects than Q85P and nonsense mutations.”

“Overall, in comparison to the other two missense mutations (Q85P, V23A), Y94N mutation seems to be more severe based on the clinical manifestations (Table 1). Therefore, we can expect nonsense mutations to be most detrimental, followed by Y94N, then Q85P, and lastly, the mildest mutation V23A.”

One conclusion that the authors make is on Lines 284-287:

“An analysis of the various nonsense and missense mutations in the PTRH2 enzyme 284 suggested that the missense mutations result in less severe phenotypes. This could be concluded from observing the less common clinical features in patients with missense mutations, as compared to the most common features.” Do the authors mean that individuals with missense changes have smaller numbers of features than individuals with nonsense mutations? Does if matter if it’s a common or less common feature? To support this the authors should present their data as stated above.

Response:

Yes, we mean that individuals with missense changes have smaller numbers of features than individuals with nonsense mutations. But since it is a relatively small number of patients and it is also a kind of neurodegenerative disease that progresses over the years, it is not possible to determine with certainty the significance of this observation, therefore larger number of patients must be followed for more than several years in order to ascertain the significance of the above observation.

Then they specifically discuss the Q85P mutation and list some “distinguishing” features though the features they list all are found in individuals with other types of mutations. They need to state the caveat of the other missense mutations is that there is only 1 patient that they base their observations in the discussion.

Response:

Yes, you are correct that the common clinical features of the mutation Q85P are also presented in other mutilations, but we specifically discussed this mutation as it has the highest relative frequency and the conclusions are based on the results from 14 affected patients as presented in tables 1 and 2.

The authors go into great lengths to discuss the possible consequences to the protein of each missense change but it seems they make a determination of which is the most deleterious by the phenotype and not so much based on the proposed functional consequences. This is likely because the protein is found in many cellular locations and has many functions and it is unknown the functions associated with each of the protein domains.

Response:

We endeavored to predict the plausible effect of various mutations to the loss in functionality of the PTRH2 protein. For example, in the discussion section, we have mentioned:

  1. “A mutation at V23 might affect, either the protein localization into the mitochondria, or the caspase-independent cell apoptosis function of the protein, or both.”
  2. “Y94 might be significant for the cell survival function of the PTHR2 enzyme.”
  3. “A Y94N mutation might have detrimental effect on interactions between PTRH2 and other proteins.”

As you suggested, PTRH2 performs several functions, many of which are not fully understood. Therefore, pinpointing the effect of mutation on a particular protein domain to a loss of a specific function based on predictive computational analysis may not be desirable. This has already been stated in the discussion section as follows:

“PTRH2 has been suggested to play a role in various cellular functions that include regulation of adhesion signals and B-cell lymphoma 2 (Bcl2) gene expression [15], modulation of Phosphoinositide 3-kinase/Protein kinase B (PI3K/AKT) and extracellular signal-regulated kinase (ERK) signaling [15], initiation of anoikis upon loss of adhesion [15,10], regulation of myogenic differentiation [1,11,16,17], regulation of mechanistic target of rapamycin (mTOR) pathway [1], and Epithelial-to-mesenchymal transition (EMT) in neural development [18,19]. These functionalities may be affected by the integrin-mediated-adhesion, binding co-factors, and phosphorylation state of the protein [15]. The effect of various PTRH2 mutations on these functional pathways remains to be elucidated. An understanding of this shall help gain insight into the development of consequent multiple manifestations observed in the IMNEPD patients.”

Other smaller issues include the following:

There are many instances that the word choices the authors use is confusing. One example is on line 37 and the word “derogatory”.

Response:

We have changed the word “derogatory” to “deleterious”.

Line 72: Does mutations in PTRH2 disrupt translation in the mitochondria or general protein translation?

Response:

As the PTRH2 protein consists of a N-terminal mitochondrial localization signal, and as also suggested by De Pereda et al. (https://doi.org/10.1074/jbc.M311449200), PTRH2 may be expected to maintain normal functioning of the translation inside the mitochondria. This has now been clarified in the manuscript as follows:

Mutations in the PTRH2 gene can lead to a reduction or loss of function of the PTRH2 protein, which may disrupt the process of mitochondrial translation and contribute to the development of IMNEPD.

It is unclear why the tables are shown in the methods section when they should be relocated to the results section. The same with the case presentation.

Response:

The tables and case presentation will be placed in the "Results" section as you suggested.

The authors discuss variable expressivity of the S43Kfs*11 mutation but it could also be said for the Q85P and W108* mutations as well. They do not discuss why this variable expressivity might occur.

Response:

This variable expressivity is valid for the other mutations also, while the phenotypic variabilities between siblings, and amongst various cases with same mutation, were also observed. And this point is mentioned in the "Discussion" section as follows: "In addition to these contrasting reports, phenotypic variabilities between siblings, and amongst various cases with same mutation, were also observed. A possible explanation for these phenotypic variabilities could be the variable expressivity of the PTRH2 gene mutations".

Once an acronym such as IMNEPD is identified, it should be used consistently throughout the manuscript. The authors define acronyms multiple times.

Response:

This was done in almost all sections of the manuscript for further clarification, but as you see it to be not necessary, so we removed them.

The results are subdivided and section 3.1 is indicated to be the bioinformatic analysis but prior to that the authors discuss the predicted protein structure which is also bioinformatically analyzed. The term bioinformatic analysis is also not very descriptive.

Response:

As you suggested, we have changed the section-heading to “Structural analysis of the mutational effects”. Also, the “predicted protein structure” section has been moved under the new heading.

Reviewer 2 Report

The manuscript consists of a case report which describes a patients affected with infantile-onset multisystem neurologic, endocrine, and pancreatic disease (IMNEPD), a synopsis of patients described so far and a bioinformatic analyses of the predicted structural consequences at the protein level of the pathogenic PTRH2 gene variants.

This is a very rare syndrome; thus, additional cases and a synthesis of current knowledge are certainly of interest.

However, the manuscript suffers from a few major weaknesses, namely the following.

- The structure of the manuscript does not comply with the common requirements: the introduction includes considerations on methods [lines 75-84]; under the material and methods section the Authors put the core results, i.e. the table and the case report.

- Methods: the Authors claim to have performed a systematic review [line 175], but there is no mention of the procedure used for literature retrieving and data synthesis, and it could not be appraised how the results were obtained.

- The clinical features of the affected individuals, including the case report, is rather confused and imprecise – e.g. ‘hand deformity’, ‘deformity of head and face’, ‘pancreatic abnormality’, etc., are not clinical signs [from the abstract throughout the whole article]. The pathological features should be analytically reported, and supported by images if possible.

In general, the terminology is often incorrect – e.g.: patients are affected from disease, not from mutations [line 141]; the presence of a genetic variants is not the expression of such variant [lines 193-195]; etc. By the way, gene names must be always in italics.

The manuscript structure, the erratic terms, coupled with the use of the English language and the overall length of the article, hamper a lean reading. As a consequence, it is difficult to pick up the main message(s) that the Authors wanted to convey.

Of note, some conclusions about the genotype-phenotype relationship were drawn in previous works [Picker-Minh 2016, Sharkia 2017, Le 2019] – the Authors should underline what original findings stem from their study.

Author Response

Reviewer-2

Thank you for your constructive review of our manuscript. All changes you requested or suggested (by you and other reviewers) had been incorporated in the manuscript and highlighted in yellow.

The manuscript consists of a case report which describes a patients affected with infantile-onset multisystem neurologic, endocrine, and pancreatic disease (IMNEPD), a synopsis of patients described so far and a bioinformatic analyses of the predicted structural consequences at the protein level of the pathogenic PTRH2 gene variants.

This is a very rare syndrome; thus, additional cases and a synthesis of current knowledge are certainly of interest.

However, the manuscript suffers from a few major weaknesses, namely the following.

- The structure of the manuscript does not comply with the common requirements: the introduction includes considerations on methods [lines 75-84]; under the material and methods section the Authors put the core results, i.e. the table and the case report.

Response:

The interference of small parts of some sections in different sections has been taken care of and removed. The case presentation and the tables have been placed in the "Results" section as you requested.

- Methods: the Authors claim to have performed a systematic review [line 175], but there is no mention of the procedure used for literature retrieving and data synthesis, and it could not be appraised how the results were obtained.

Response:

We performed a comprehensive review of literature on all IMNEPD cases studied so far, as we conducted a thorough search in PubMed database and Google scholar as well as a local database of rare genetic disorders from Israel. Additionally, to increase the confidence in the in silico prediction results, we endeavored to retrieve accurate structural and sequence data from widely used databases, such as Protein Data Bank (PDB) and UniProt. Further, we use popularly cited webservers based on well-accepted stringent algorithms, such as AlphaFold, ConSurf, Rosetta-PM etc., to present reliable prediction results.

- The clinical features of the affected individuals, including the case report, is rather confused and imprecise – e.g. ‘hand deformity’, ‘deformity of head and face’, ‘pancreatic abnormality’, etc., are not clinical signs [from the abstract throughout the whole article]. The pathological features should be analytically reported, and supported by images if possible.

Response:

As our study reviews the available cases of IMNEPD from published research articles, therefore, we depended on the exact descriptions made by the authors themselves, particularly, regarding the clinical signs and phenotypic features. 

In general, the terminology is often incorrect – e.g.: patients are affected from disease, not from mutations [line 141]; the presence of a genetic variants is not the expression of such variant [lines 193-195]; etc. By the way, gene names must be always in italics.

Response:

We checked the manuscript and corrected the terminology as requested noting that PTRH2 is italicized when it refers to the gene, while when it refers to the protein it remains normal fonts. The other corrections incorporated in the manuscript are as follows:

Line 141

The routine genetic evaluation including CMA (Chromosomal microarray analysis) was carried out and found to be normal (no chromosomal microarray aberrations were detected). Since the clinical manifestations of the proband, resemble those of IMNEPD, sequencing for PTRH2 gene was performed. The results showed that the patient is homozygous for the missense mutation p. Gln85Pro (c.254A>C) which is the variant we previously identified [3] and found to be a pathogenic variant in PTRH2 gene.

Lines 193-195

There are four different nonsense mutations in PTRH2 gene. Three of them (A90Gfs*13, W108*, and E110*) were found in one family each. While the forth one (S43Kfs*11) had been found in two unrelated families (Table 1).

The manuscript structure, the erratic terms, coupled with the use of the English language and the overall length of the article, hamper a lean reading. As a consequence, it is difficult to pick up the main message(s) that the Authors wanted to convey.

Response:

We appreciate your comment, however, we've done our best to clarify any confusion present in any part of the manuscript and added many portions that enhanced the process of reading and understanding. These changes had been highlighted in yellow.

Of note, some conclusions about the genotype-phenotype relationship were drawn in previous works [Picker-Minh 2016, Sharkia 2017, Le 2019] – the Authors should underline what original findings stem from their study.

Response:

In addition to reporting a new case with a previously documented mutation, the structural analysis of the mutational effects, and their plausible effect on the protein function, presents novelty to our manuscript. We have mentioned the same in the discussion section now:

“In this manuscript, we report a new case with a previously documented mutation. Further, we have endeavored to enlist the previously reported phenotypic symptoms for each (missense and nonsense) condition, and provide a computational insight into the plausible reasons for the severity of the observed symptoms. In this direction, we performed a structural analysis to understand the effect of various mutations (missense and nonsense) on the intra-molecular stability and overall conformation of the protein.”

Reviewer 3 Report

PTRH2 gene variants: Recent review of the phenotypic features 2 and their bioinformatics analysis

A brief summary

This review manuscript aimed to study the variant of Peptidyl-tRNA hydrolase 2 (PTRH2)

gene in rare autosomal recessive disorder characterized by an infantile-onset multisystem neurologic, endocrine, and pancreatic disease (IMNEPD). Also, the authors study the bioinformatics analyses on the some of these variants.

Specific comments

1. It would be better if all the tables were redesigned because they look too busy and readers might get lost

Author Response

Reviewer-3

Thank you for your constructive review of our manuscript. All changes you requested or suggested (by you and other reviewers) had been incorporated in the manuscript and highlighted in yellow.

A brief summary

This review manuscript aimed to study the variant of Peptidyl-tRNA hydrolase 2 (PTRH2)

gene in rare autosomal recessive disorder characterized by an infantile-onset multisystem neurologic, endocrine, and pancreatic disease (IMNEPD). Also, the authors study the bioinformatics analyses on the some of these variants.

Thank you for your comment, we highly appreciate your cooperation.

Specific comments

  1. It would be better if all the tables were redesigned because they look too busy and readers might get lost

Response:

The tables had been redesigned and modified as you requested.

Round 2

Reviewer 1 Report

The representation of the data is much improved as well as the clarity of the discussion. Some awkward wording still exists.

Author Response

Dear Reviewer 1

We again thank you for your efforts. We did our best to enhance the quality of our manuscript by clarifying and modifying many terms and sentences that seemed to be previously unclear, based on your suggestion. The changes that we incorporated are marked using the "track change" function of MS Word as recommended by the editor.

Now, we hope that our manuscript is suitable for publication.

With kind regards,

Prof. Sharkia and the research team 
